# LPA_1_ Receptor Promotes Progesterone Receptor Phosphorylation through PKCα in Human Glioblastoma Cells

**DOI:** 10.3390/cells10040807

**Published:** 2021-04-04

**Authors:** Silvia Anahi Valdés-Rives, Denisse Arcos-Montoya, Marisol de la Fuente-Granada, Carmen J. Zamora-Sánchez, Luis Enrique Arias-Romero, Olga Villamar-Cruz, Ignacio Camacho-Arroyo, Sonia M. Pérez-Tapia, Aliesha González-Arenas

**Affiliations:** 1Departamento de Medicina Genómica y Toxicología Ambiental, Instituto de Investigaciones Biomédicas, Universidad Nacional Autónoma de México (UNAM), 04510 Ciudad de México, Mexico; anahiivaldes@gmail.com (S.A.V.-R.); diam.denisse@gmail.com (D.A.-M.); mdelafuente@iibiomedicas.unam.mx (M.d.l.F.-G.); 2Unidad de Investigación en Reproducción Humana, Instituto Nacional de Perinatología-Facultad de Química, Universidad Nacional Autónoma de México (UNAM), 04510 Ciudad de México, Mexico; carmenjaninzamora@comunidad.unam.mx (C.J.Z.-S.); camachoarroyo@gmail.com (I.C.-A.); 3Unidad de Investigación en Biomedicina (UBIMED), Facultad de Estudios Superiores-Iztacala, Universidad Nacional Autónoma de México (UNAM), Tlalnepantla, 54090 Estado de México, Mexico; vicovc@gmail.com (O.V.-C.); larias@unam.mx (L.E.A.-R.); 4Unidad de Desarrollo e Investigación en Bioprocesos (UDIBI), Escuela Nacional de Ciencias Biológicas, Instituto Politécnico Nacional, 11350 Ciudad de México, Mexico; sperezt@ipn.mx; 5Departamento de Inmunología, Escuela Nacional de Ciencias Biológicas, Instituto Politécnico Nacional, 11340 Ciudad de México, Mexico

**Keywords:** glioblastoma, LPA_1_ receptor, protein kinase C α, progesterone receptor

## Abstract

Lysophosphatidic acid (LPA) induces a wide range of cellular processes and its signaling is increased in several cancers including glioblastoma (GBM), a high-grade astrocytoma, which is the most common malignant brain tumor. LPA_1_ receptor is expressed in GBM cells and its signaling pathways activate protein kinases C (PKCs). A downstream target of PKC, involved in GBM progression, is the intracellular progesterone receptor (PR), which can be phosphorylated by this enzyme, increasing its transcriptional activity. Interestingly, in GBM cells, PKCα isotype translocates to the nucleus after LPA stimulation, resulting in an increase in PR phosphorylation. In this study, we determined that LPA_1_ receptor activation induces protein-protein interaction between PKCα and PR in human GBM cells; this interaction increased PR phosphorylation in serine400. Moreover, LPA treatment augmented *VEGF* transcription, a known PR target. This effect was blocked by the PR selective modulator RU486; also, the activation of LPA_1_/PR signaling promoted migration of GBM cells. Interestingly, using TCGA data base, we found that mRNA expression of *LPAR1* increases according to tumor malignancy and correlates with a lower survival in grade III astrocytomas. These results suggest that LPA_1_/PR pathway regulates GBM progression.

## 1. Introduction

Glioblastoma (GBM), an astrocytoma grade IV, represents the maximal evolution grade of astrocytomas and it is among the most lethal human malignancies [1,2]. The median survival for GBM patients with the best therapy is 12–15 months and only a minor percentage of the patients (3–5%) survive for more than three years. In the Mexican population, it has been estimated that the mean age of incidence is 45 ± 15 years, compared to the global estimate of 60 ± 15 years [3,4].

The lysophosphatidic acid (LPA) is a small molecule with a phosphate head group and an acyl (or alkyl) chain at the sn-1 (or sn-2) position of the glycerol backbone [5]. It interacts with at least six GPCRs (i.e., LPA_1–6_) coupled to the Gα proteins: Gαq/11, Gαi/0, Gα12/13, Gαs; LPA can exert an extensive range of physiological effects such as wound healing, proliferation, neurogenesis, angiogenesis and survival, depending on the cellular context [6,7]. LPA is mainly synthesized by the cleavage of membrane phospholipids into lysophospholipids by removing a fatty acid chain through phospholipase A (PLA1 or PLA2) [8]. Subsequently, autotaxin (ATX) cleaves the head group (e.g., choline, ethanolamine, or serine) on the lysophospholipids and turns them into LPA [8,9]. ATX (also known as *ENPP2*) is a 125 kDa enzyme from the family of ectonucleotide pyrophosphatases/phosphodiesterases. This enzyme is secreted to the extracellular space in a constitutively activated form and is the responsible for the main LPA production in many pathologies [8,10].

Aberrant LPA signaling has been linked to diverse conditions such as cancer [8,11]; besides, the high expression of the LPA_1_ receptor has been associated with malignant progression by enhancing proliferation, migration, angiogenesis and cancer stem-cell maintenance in several tumors such as breast cancer, pancreatic cancer, ovarian cancer and GBM [12,13,14,15]. In GBM cells, the LPA_1_ receptor is redistributed in the cell membrane, increasing its coupling to Gαq and Gα12 proteins. This activates protein kinases C (PKCs) [16,17] and turns on a signaling pathway to induce malignant progression [17,18,19,20].

The role of PKCs in cancer progression is well-known [21]. In GBM, the kinase with the highest expression is PKCα, [16,22]. This PKC isotype induces a pro-survival and proliferative effects in GBM cells [23,24]; however, since this kinase has a wide range of actions, its contribution to GBM progression through specific signaling pathways is poorly understood [21]. We have previously demonstrated that LPA, through its LPA_1_ receptor, promotes PKCα nuclear translocation, inducing progesterone receptor (PR) phosphorylation at S400 [25].

In previous work, we have demonstrated that activation of PKCs phosphorylates PR at S400 and that this phosphorylation induces PR transcriptional activity [24,25]. The activation of PR is known to increase proliferation, migration and invasion of GBM cells [26,27,28] through the transcription of essential factors for tumor growth, such as *VEGF*, *cyclin D1* and *EGFR* [29]; however, it is unknown whether PR phosphorylation by PKC is through a direct or an indirect interaction and if this posttranslational modification results in activation of PR targets transcription.

Therefore, this study aimed to evaluate if LPA induces PKCα/PR interaction and if this association phosphorylates and activates PR, thereby increasing GBM cell migration. Using the TCGA database, we correlated patient survival with mRNA expression of *LPAR1*, *LPAR3* and *ENPP2* genes.

## 2. Materials and Methods

### 2.1. Cell Culture and Treatments

Human glioblastoma-derived cell lines U251 and LN229 (American Type Culture Collection, Manassas, VA, USA (ATCC)) were grown in 10-cm dishes and maintained in DMEM medium (In Vitro, CDMX, Mexico), supplemented with 10% fetal bovine serum at 37 °C under a 95% air, 5% CO2 atmosphere. 1-Oleoyl Lysophosphatidic Acid (LPA; 62215; Cayman Chemical, Ann Arbor, MI, USA) was used to activate LPA receptors and subsequently PKCα. LPA_1/3_ receptor antagonist Ki16425 (SML0971; Sigma-Aldrich, St. Louis, MO, USA) was added 30 min before the LPA treatment when used. Progesterone (P4) (SML0971; Sigma-Aldrich, St. Louis, MO, USA) was used to activate PR; selective PR modulator, RU486 (M8046; Sigma-Aldrich, St. Louis, MO, USA), was added 30 min before the P4 treatment when used.

The National Institute of Genomic Medicine in Mexico City did proof of cell validation for the U251 cell line and for LN229, we get the ATCC certificate analysis.

### 2.2. Proximity Ligation Assay

PLA experiments were performed using Duolink^®^ kit (DUO92101, Sigma-Aldrich, St. Louis, MO, USA) according to the manufacturer’s instructions. A total of 4000 cells were plated in 16 well chambers (178599, Nunc Lab-Teck, Thermo Scientific, Waltham, MA, USA) in DMEM medium with 10% fetal bovine serum for 24 h. Eighteen hours before treatments, the medium was changed for phenol red-free DMEM without fetal bovine serum and cells were incubated at 37 °C under a 95% air and 5% CO_2_ atmosphere. Afterward, cells were stimulated for 0, 5 and 15 min with LPA 100 nM, washed twice with ice-cold PBS, fixed for 15 min with PBS/Paraformaldehyde 4% and permeabilized with 0.5% Triton X−100 for 30 min. Then, cells were blocked with 40 μL of blocking solution for 1 h at 37 °C in a humidity chamber and incubated overnight at 4 °C with the primary antibodies: monoclonal mouse antibody against PKCα (2 µg/mL; sc-8393; Santa Cruz Biotechnology, Dallas, TX, USA) and polyclonal rabbit antibody against total Progesterone Receptor (2 µg/mL; sc-7208; Santa Cruz Biotechnology, Dallas, TX, USA). To detect the primary antibodies, secondary proximity probes binding rabbit and mouse immunoglobulin (PLA probe rabbit PLUS and PLA probe mouse MINUS, Olink Bioscience, Sigma-Aldrich, St. Louis, MO, USA) were diluted 1:15 and 1:5 in blocking solution, respectively. The cells were then incubated with the proximity probe solution for 1 h at 37 °C, washed three times in 50 mM Tris pH 7.6, 150 mM NaCl, 0.05% Tween-20 (TBS-T) and incubated with the hybridization solution containing connector oligonucleotides (Olink Bioscience, Sigma-Aldrich, St. Louis, MO, USA) for 45 min at 37 °C. Samples were washed with TBS-T and subsequently incubated in the ligation solution for 45 min at 37 °C. The ligation solution contained T4 DNA ligase (Fermentas, Sigma-Aldrich, St. Louis, MO, USA), allowing the ligation of secondary proximity probes and connector oligonucleotides to form a circular DNA strand. Subsequently, the samples were washed in TBS-T and incubated with the amplification solution, containing phi29 DNA polymerase (Fermentas, Sigma-Aldrich, St. Louis, MO, USA) for the roller cycle amplification for 90 min at 37 °C and washed three times with TBS-T. Finally, the samples were incubated with the detection mix solution (containing Texas Red-labeled detection probes that recognize the amplified product, Olink Bioscience, Sigma-Aldrich, St. Louis, MO, USA) for 1 h at 37 °C, washed twice in SSC-T buffer (150 mM NaCl, 15 mM sodium citrate, 0.05% Tween-20, pH 7) and were coverslipped with a fluorescence mounting medium (Biocare Medical, Pacheco, CA, USA). Fluorescent signals were detected by laser scanning microscopy (Nikon TE2000, Amsterdam, Netherlands) and PLA-positive signals were quantified using MetaMorph software. At least 50 nuclei were measured for each experimental condition.

### 2.3. RNA Extraction and RT-qPCR

To evaluate the effect of LPA on the gene expression induced by PR, we quantified VEGF, EGFR, TGFβ1 and cyclin D1 expression 24 h after LPA or P4 stimulation. 2 × 10^5^ U251 cells were plated per well in six-well plates. At 24 h before treatment, the growth medium was replaced by phenol red-free DMEM supplemented with 10% charcoal-dextran filtered SFB. Then, cells were treated for 24 h with LPA (100 nM), P4 (10 nM), RU 486 (1 µM) and the conjunct treatments LPA + RU486 or P4 + RU486 (same concentrations). After treatment, cells were washed with PBS and TRIzol^®^ LS Reagent (Invitrogen, Carlsbad, CA, USA) was added to detach cells from the plate. The RNA extraction was performed according to the manufacturer recommendations by the phenol-guanidine isothiocyanate-chloroform method. After extraction, RNA quantity and purity were measured with the NanoDrop 2000 Spectrophotometer (Thermo Fisher Scientific, Waltham, MA, USA). Additionally, optimal RNA integrity was assessed by the electrophoresis of 1 µg of total RNA in a 1% agarose gel in 0.5× TB buffer. cDNA was synthesized from 1 µg of total RNA with the M-MLV Reverse Transcriptase (Invitrogen, Carlsbad, CA, USA) and oligonucleotides (dT)12–18 Primer (Invitrogen, Carlsbad, CA, USA) according to the manufacturer’s protocol. A total of 2 µL of such reaction was used to determine the expression of VEGF, EGFR, or the 18S ribosomal gene as expression control by qPCR with the LightCycler^®^ FastStar DNA Master SYBR Green I and the LightCycler 1.5 (Roche Molecular Systems, Pleasanton, CA, USA) according to the manufacturer’s protocol. Oligonucleotide sequences were: 5′-CCACACCATCACCATCGACA-3′ forward VEGF primer, 5′-CCAATTCCAAGAGGGACCGT-3′ reverse VEGF primer (amplified fragment of 153 bp); 5′-GCCTTGACTGAGGACAGGCAT-3′ forward EGFR primer, 5′-TGGTAGTGTGGGTCTCTGCT-3′ reverse EGFR primer (amplified fragment of 152 bp); 5′-AGTGAAACTGCAATGGCTC-3′ forward 18S primer, 5′-CTGACCGGGTTGGTTTTGAT-3′ reverse 18S primer (amplified fragment of 167 bp). A reaction without RT was performed as a negative control. The relative gene expression was calculated with the 2^−ΔCt^ [30,31]. One-way ANOVA and a Tukey test were performed to determine statistical differences between treatments by using the GraphPad Prism 8.0 software (GraphPad Software, San Diego, CA, USA).

### 2.4. Immunofluorescence

Eight thousand cells per well were plated in Millicell EZ 4-well glass slides (Millipore, Burlington, MA, USA). Eighteen hours before treatments, the medium was changed for phenol red-free DMEM without fetal bovine serum and cells were incubated at 37 °C under a 95% air, 5% CO_2_ atmosphere. After the treatments, cells were fixed for 20 min in 4% paraformaldehyde solution at 37 °C and permeabilized with 100% methanol for 6 min at −4 °C. Next, fixed cells were blocked with 1% bovine serum albumin in PBS for 1 h at room temperature and incubated at 4 °C for 24 h with 1µg/mL of rabbit anti-pS400PR (ab60954, Abcam, Cambridge, UK) in 0.5% bovine serum albumin in PBS. The samples were rinsed thrice in PBS for 5 min each and incubated in the dark with 0.5 µg/mL anti-mouse Alexa Fluor 488-labeled secondary antibodies (A11034, Invitrogen, Carlsbad, CA, USA) for 45 min. Nuclei were stained with 1 µg/mL Hoechst 33342 solution (Thermo Scientific, Waltham, MA, USA). The cells were coverslipped with a fluorescence mounting medium (Biocare Medical, Pacheco, CA, USA). The samples were visualized in a Nikon A1R + STORM confocal microscope.

Specific characteristics of the antibodies described in Section 2.2 and Section 2.4 can be consulted in Appendix A.

### 2.5. Migration Assay

To evaluate if blocking LPA_1_/PR signaling pathway affects migration, the scratch-wound assay was used. 300,000 cells were plated with DMEM medium in six-well plates until 90% confluence and the formation of a uniform monolayer were observed. With a 200 mL pipet tip, a scratch per well was made in a previously marked well to allow the identification of 4 separate segments of scratch. 

The detached cells were washed by aspiration. Cells were incubated with DMEM medium with DNA synthesis inhibitor cytosine β-D-arabinose-furanoside hydrochloride (10μM, AraC; Sigma-Aldrich, St. Louis, MO, USA). One hour later, RU486 and Ki16425 were added and the scratch images were taken with an inverted microscope Olympus IX71 (Olympus Corporation, Shinjuku City, Tokyo, Japan), at 4X magnification at 0 and 24 h. The percentage of wound closure from migrating cells in the scratch area was calculated as the mean of four fields using the ImageJ software (National Institutes of Health, Bethesda, MD, USA) and the formula:(1)$$Wound\;closure\;\%\;=\;\left[\frac{At0h\;-\;At\Delta h}{At0h}\right]\times 100$$
where: *At*0*h* = total area at time 0; *At*Δ*h* = total area after “×” hours.

### 2.6. LPAR1/3 and ENPP2 Gene Expression Evaluation

For *LPAR1*, *LPAR3* and *ENPP2* (ATX gene) mRNA determination, we used the California University, Santa Cruz platform: UCSC Xena (XenaBrowser.net; accessed on 18 March 2021) and the database from TCGA. After applying the filters and deleting the duplicated ones, we obtained 258 samples for grade II (GII), 271 for grade III (GIII) and 166 for grade IV astrocytomas (GIV or GBM). mRNA levels were obtained and plotted on Graph Pad Prism 5.0.

### 2.7. Survival Curves

For the evaluation of *LPAR1*, *LPAR1* and *ENPP2* (ATX gene) mRNA relationship with patient survival, Kaplan–Meier curves were performed using the platform of the California University, Santa Cruz: UCSC Xena (XenaBrowser.net; accessed on 18 March 2021) and the database from TCGA for tumor tissue. We obtained the following data from “TCGA low-grade astrocytoma and glioblastoma” and after applying the filters and deleting the duplicated ones: for *LPAR1*: Grade II: 65 samples, Grade III: 132 samples and Grade IV: 166 samples. For *LPAR3*: Grade II: 129 samples, Grade III: 134 samples and Grade IV: 96 samples. For *ENPP2*: Grade II: 254 samples, Grade III: 265 samples and Grade IV: 153 samples. Statistical analysis between high and low expression was performed with a Log-rank (Mantel–Cox) test. A value *p* < 0.05 was considered statistically significant, as stated in figure legends. All these analyses were performed in GraphPad Prism 8.0 (Graph Pad Software, San Diego, CA, USA).

### 2.8. Spearman Correlations

To obtain the correlation between the degree of expression of *PGR* and *LPAR1* or *ENPP2*, the gene expression database called “TCGA Glioblastoma (GBM)” was extracted from which the samples that did not have data for both genes were eliminated; 155 samples were used for each graph and the Spearman coefficient was determined using the GraphPad Prism 8.0.2 program.

### 2.9. Statistical Analysis

All data were analyzed and plotted using the GraphPad Prism 5.0 software for Windows XP (GraphPad Software, San Diego, CA, USA). Statistical analysis of comparable groups was performed using a one-way ANOVA with a Bonferroni or Tukey’s post-test. A value of *p* = 0.05 or less was considered statistically significant, as stated in figure legends.

## 3. Results

### 3.1. PKCα Interacts with PR after LPA_1_ Receptor Activation

Our previous work showed that PKCα immunoprecipitated with total and phosphorylated PR at residue S400 [25]; we have also demonstrated that LPA induces PKCα nuclear translocation [26]. To investigate if these proteins could interact in the same compartment after LPA_1_ receptor activation, a Proximity Ligation Assay (PLA) was performed (Figure 1).

Our results show that in the U251 cell line, the stimulation with LPA at 5 and 15 min induced a protein-protein interaction between PKCα and PR with a peak at 15 min. Additionally, the puncta were located both in the cytoplasmatic and the nuclear compartments. Interestingly, a scarce basal interaction of both proteins was observed.

### 3.2. PR Phosphorylated in Serine 400 (PRpS400) Has a Nuclear Localization

We have previously demonstrated LPA stimulation-induced PRpS400 and PKCα nuclear translocation [25]. We were interested in knowing if PRpS400 was cytoplasmic or/and nuclear since PKCα and PR interaction occurred in both compartments.

To assess the phosphorylation of PRpS400, we used immunofluorescence after 15 min of stimulation with 100 nM of LPA and/or 2.5 μM of Ki16425 in U251 and LN229 cell lines (Figure 2).

Interestingly, our results show nuclear puncta of PRpS400 in both cell lines at 15 min after LPA stimulation. We measured the signal at 15 min because we have previously demonstrated that PRpS400 reaches its maximum level at this timepoint [25].

In contrast, Ki16425, an inhibitor of LPA_1/3_ receptors, blocked this effect on both cell lines. We have previously demonstrated that U251 cells express the LPA_1_ receptor at a protein level and LN229 cells express LPA_1_/LPA_3_ receptors [25]. In the LN229 cell line, PRpS400 could be promoted by both LPA receptors; meanwhile, in U251, this phosphorylation is induced through the LPA_1_ receptor.

### 3.3. PR Induces the Expression of VEGF after Its Activation via the LPA_1_ Receptor

In previous work, we showed that activating the LPA_1_ receptor with its ligand induces PKCα translocation to the nucleus at 5 and 15 min [26]. Since this kinase interacts with PR and induces its phosphorylation at S400 [25], we wanted to test whether PR gene targets could be induced by LPA treatment. We chose two important PR (*VEGF* and *EGFR*) targets known to contribute to GBM progression.

To evaluate whether LPA receptors activation modulates the expression of PR target genes, we measured their respective mRNAs with RT-qPCR after 24 h of stimulation with 10 nM of P4, 100 nM of LPA, 1 μM of RU486 (selective modulator of PR) and the combination of RU486 with P4 or LPA in U251 cell line (Figure 3).

The results show that LPA modulated *VEGF* expression, similarly to P4, after 24 h of stimulation. The mRNA expression was downregulated when stimuli were combined with the selective PR modulator, RU486. However, this modulator acted as a receptor agonist when used alone. *EGFR* mRNA expression increased after P4 treatment and RU486 blocked this effect; however, no regulation by LPA was observed (Figure 3B).

### 3.4. LPA/PR Pathway Induces Migration

VEGF, a PR downstream target, is known to promote tumor progression. Migration of cancer cells and new vessel formation are vital factors in sustaining a growing tumor [32]. In GBM, a highly vascular tumor, VEGF is an important growth factor known to induce migration of both: GBM and endothelial cells [33].

We were interested in assessing if the LPA/PR pathway could induce migration in U251 and LN229 cell lines. We evaluated this process through a wound-healing assay with 10% FBS as LPA source combined with Ki1645 and/or RU486 at 0 h and 24 h (Figure 4).

The percentage of wound closure without or after treatment with DMSO or EtOH was 85% for U251 and 70% for LN229. In the U251 cell line, the effect of Ki16425 and RU486, in wound closure was 60 and 65%, respectively; meanwhile, in LN229 was 35% with Ki16425 and 49% with RU486. Interestingly, both compounds (RU486 and Ki16425) caused a 55% closure in U251 and a 30% in LN229, suggesting a slightly additive effect of both inhibitors.

Ki16425 is an antagonist for LPA_1/3_ receptors. We have previously demonstrated that U251 cells express the LPA_1_ receptor at a protein level and LN229 cells express LPA_1_/LPA_3_ receptors [25]; thus, in the LN229 cell line, inhibition of migration could be promoted by both LPA receptors.

### 3.5. LPA_1_, LPA_3_ and ATX Expression in the Survival of GBM Patients

Our results showed that LPA_1_/PR pathway induced PR targets transcription and migration of GBM cells; therefore, it is probable that this signaling pathway enhances tumor progression. Thus, we questioned whether LPA receptors (*LPAR1*, Figure 5A; *LPAR3*, Figure 6A) or ATX (*ENPP2*; Figure 7A) expression is modified at mRNA level in different astrocytoma grades. Additionally, we were interested in studying the correlation between these genes’ expression and patient survival. Therefore, we constructed Kaplan–Meier graphs using TCGA data for astrocytoma grades II, III and IV with low and high expression of *LPAR1* (Figure 5B–D), *LPAR3* (Figure 6B–D) and *ENPP2* (Figure 7B–D). 

These results showed that the expression of *LPAR1* among the different grades of astrocytomas is increased with tumor malignancy (Figure 5A) while *LPAR3* and *ENPP2* expression does not (Figure 6A and Figure 7A); in fact, *ENPP2* expression decreases according to tumor malignancy. For Kaplan–Meier curves, we found that *LPAR1* low expression is the only one related to overall survival in astrocytoma grade III patients (Figure 5C).

### 3.6. Correlation between PGR and ENPP2 Gene Expression

Although *LPAR1* mRNA expression was associated with poor prognosis only in grade III astrocytomas, we were interested in evaluating whether there was a correlation between *PGR* and *LPAR1* expression (Figure 8A).

Our results show that there was no correlation between the expression of *LPAR1* and *PGR* genes. Since LPA production is mainly due to the enzyme ATX [8,9,10], it is possible that the expression of this enzyme could impact the LPA_1_/PR signaling pathway. Therefore, we also analyzed if there was a correlation between *ENPP2* (ATX gene) and *PGR* mRNA expression. In this case, there was a positive correlation between these genes’ expression (Figure 8B).

Additionally, by using PROMO-ALGGEN software (http://alggen.lsi.upc.es/; accessed on 5 March 2021), we investigated if *ENPP2* had possible progesterone response elements (PRE) in its promoter (*ENPP2* promoter sequence was obtained at the Eukaryotic Promoter Database (https://epd.epfl.ch//index.php; accessed on 5 March 2021) for PR binding that could, further, explain their correlation. We found seven possible binding sites in the *ENPP2* promoter with over 90% probability and one with over 95% (Figure 9) for both PR receptor isoforms (PRA and PRB).

## 4. Discussion

In this work, using the PLA assay, we evaluated whether there is a protein-protein interaction between PR and PKCα. The in situ PLA assay identifies the physical proximity of proteins; a signal will only be produced if two proteins are at a distance of less than 40 nm; this provides high selectivity for protein–protein interaction detection [34].

This assay demonstrated that after 5 and 15 min of 100 nM LPA stimulation, the proximity between PR and PKCα increased in a time-dependent manner. This result suggests a direct interaction of the kinase with the receptor that leads to the latter’s phosphorylation. It is interesting to point out that U251 cells reached a high level of interaction between PR and PKCα at 15 min after LPA treatment, at the same time in which the highest accumulation of PKCα in the nucleus and the phosphorylation peak of PRpS400 were observed [25]. Furthermore, the protein-protein interaction location was cytoplasmatic and nuclear; this result proves that the PKCα/PR association could occur in both compartments, although the accumulation of PRpS400 was mainly nuclear.

Additionally, we wanted to evaluate whether the LPA_1_/PKCα/PRpS400 pathway could modulate PR target genes’ expression. Previous work from our laboratory reported that activation of PKCα increased PR phosphorylation that induced its transcriptional activity. In turn, PR increased the mRNA expression of the blocking factor induced by progesterone (PIBF) [27], a known PR target gene that causes cellular proliferation of GBM [24].

Other genes involved in GBM cell proliferation and migration are *EGFR* and *VEGF*, which are regulated by PR [29]. Therefore, we analyzed whether PR activation by LPA could modulate the expression of these two genes. We observed that *VEGF* expression was upregulated after LPA treatment similarly as P4 did it. In contrast, *EGFR* expression was only increased when the cells were treated with P4.

*EGFR* and *VEGF* are differentially regulated. *VEGF* has PRE consensus sites in addition to AP1 and SP1 sites in its promoter [35,36]. In previous work, we demonstrated that PKC activation induced a two-fold increase in PR transcriptional activity when GBM cells (U251) were transfected with the MMTV-Luc reporter plasmid carrying two PRE [24]; thus, LPA treatment induces PKCα activation (through LPA_1_ receptor), that in turns phosphorylates PR, modulating *VEGF* expression by PRE.

Conversely, *EGFR* has SP1 sites but does not have canonical PREs. It is known that PR can activate MAPK or c-Src after its stimulation with P4 and that these signaling cascades promote receptor phosphorylations in different sites. Phosphorylations can promote PR binding to SP1 sites (e.g., PRpS345 by MAPK), inducing the transcription of genes without PRE in their promoters [37,38,39], which also explains the lack of RU486 effects on *EGFR* expression.

Although the selective modulator RU486 combined with P4 and LPA inhibits *VEFG* expression, we observed an increase in this gene transcription when used alone. This may be due to the role of RU486 as a partial agonist [40,41].

In addition, RU486 is a modulator of the glucocorticoid receptor, which could also induce *VEGF* transcription [42].

Additionally, the wound healing assay showed that inhibition of LPA_1_/PR reduces cell motility. Interestingly, the inhibitors’ combined effect had a slight additive impact suggesting that other pathways such as MAPK could contribute to cell migration through LPA_1_ receptor signaling cascade or non-canonical PR pathways. It is worth noting that despite using a complete medium as a source of LPA, which includes growth factors, the effects of inhibiting LPA_1_/PR or both were marked. Thus, due to the invasiveness nature of GBM, these results suggest that targeting both proteins is potentially beneficial at a therapeutical level. Moreover, the signaling from both proteins impacts more than just migration; although further studies are needed to corroborate our findings, they could also affect invasion, proliferation, survival and other important tumor progression features. It would be essential to point out that U251 cells express the LPA_1_ receptor at a protein level and LN229 cells express LPA_1_/LPA_3_ receptors [25]; thus, the results found in the LN229 cell line could be promoted by both LPA receptors; meanwhile, in U251, LPA_1_ receptor would be the main receptor involved in PR phosphorylation by PKC and in promoting cell migration.

We were also interested in analyzing if the mRNA of these LPA receptors and autotaxin could function as a predictor of patient survival among astrocytomas. Remarkably, *LPAR1* increases with tumor malignancy. However, it was only linked to survival in astrocytomas grade III. *LPAR3* mRNA expression did not vary among grades and *ENPP2* (ATX gene) decreased with malignancy; neither mRNAs had a link to patient prognosis. It is worth noting that the TCGA database does not harbor data of mutations of LPA receptor or ATX; therefore, we cannot account for mutants derived from *LPAR1, LPAR3* and *ENPP2* that could impact patient survival.

Additionally, GBM differences between the transcriptome and proteome have been found, suggesting that mRNA levels do not always reflect protein content. Lemeé et al. found a poor correlation between mRNA and protein content of the neurofilament light polypeptide and synapsin 1 in GMB patient samples [43]. Moreover, Song et al. performed a proteomic and genomic profiling of GBM and normal brain tissue and found mRNA expression rarely correlated with protein content [44]. The latter suggests that despite de levels of *LPAR3* and *ENPP2* gene expression, we cannot disregard their protein role in GBM progression. In contrast, it would seem that *LPAR1* mRNA levels could be correlated with high protein levels, as demonstrated previously [25] and it has a functional role in tumor progression. However, none of the mRNA evaluated in this work could be markers for survival prognosis in GBM.

Furthermore, in a recent work from our lab, we have shown that PR protein expression increases with tumor malignancy, although mRNA levels seemed to decrease with tumor grade [45]. Moreover, PR mRNA does not correlate with survival prognosis in GBM. Several critical factors implicated in mRNA translation into its protein can be altered in GBM (e.g., regulatory proteins and siRNAs that need to be investigated [46]; therefore, the causes of mRNA/protein inconsistency in GBM remain unknown.

Despite the latter, our data suggested a possible correlation of *ENPP2* regulation by *PGR*, which could be through a positive feedback loop since we found probable PRE in the *ENPP2* promoter; thus, PRpS400 through LPA_1_ receptor activation may upregulate ATX in GBM. However, the evidence so far is insufficient and requires further investigation.

## 5. Conclusions

LPA stimulates PKCα-PR interaction inducing receptor phosphorylation at S400, mainly in the cell nucleus. LPA_1_/PR signaling cascade upregulates *VEGF* expression and induces cell migration contributing to GBM progression.

The expression of *LPAR3* and *ENPP2* mRNAs is not a biomarker for prognosis in patients’ survival.

## Figures and Tables

**Figure 1 cells-10-00807-f001:**
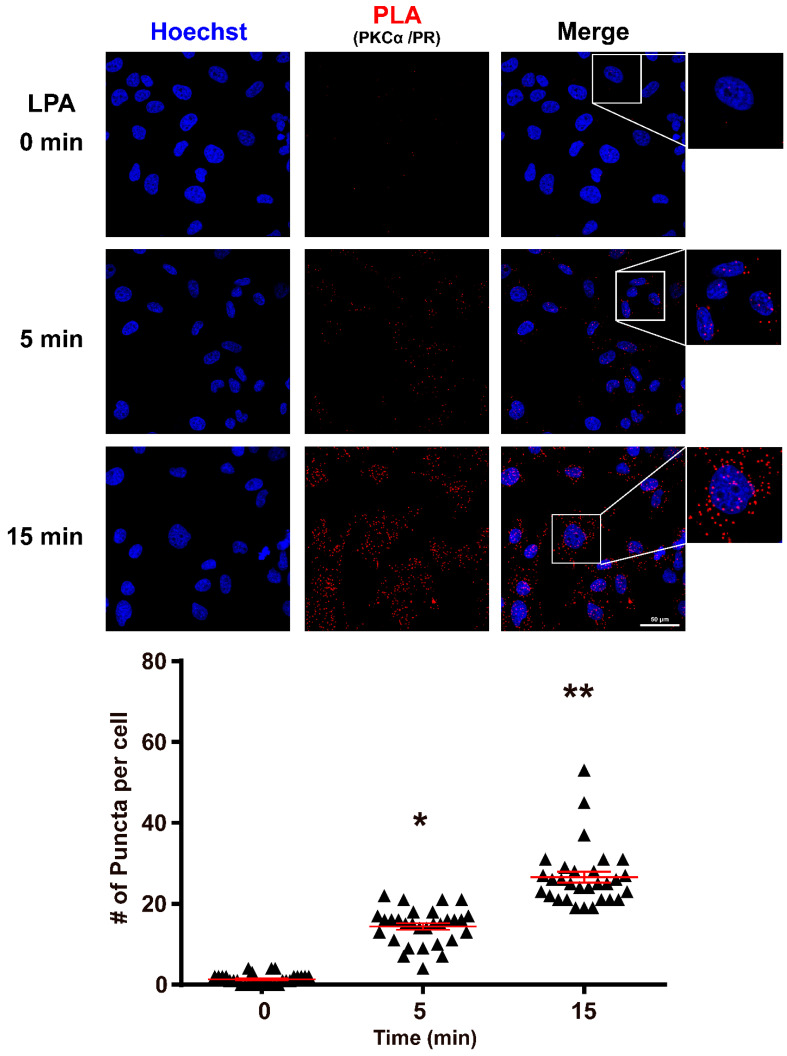
LPA induces the interaction of PKCα and PR. Top panel: representative images of the PLA assay. The red dots denote the proximity (distance less than 40 nm) of PKCα and PR at 0, 5 and 15 min of stimulation with 100 nM of LPA in the U251 cell line. Lower panel: graphic representation of the interactions. Results are expressed as the mean ± S.E.M. of 30 cells nuclei per condition. A one-way ANOVA statistical test followed by a Tukey’s post-test was used. * *p* < 0.001, compared to control (0) and 15 min, ** *p* < 0.001 compared to control (0).

**Figure 2 cells-10-00807-f002:**
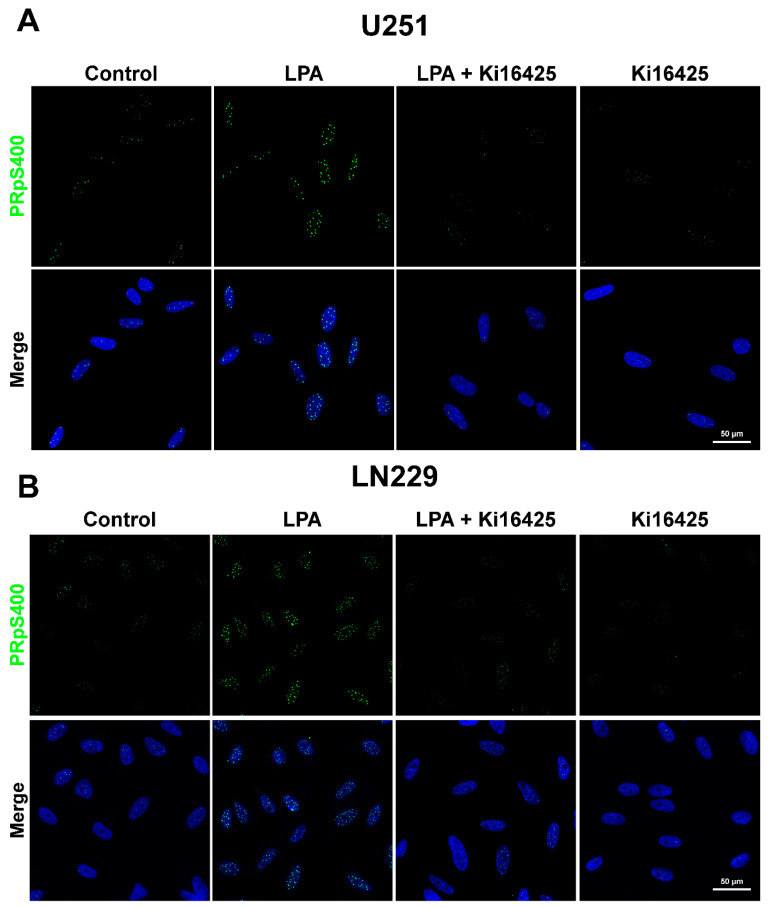
Localization of PRpS400 in GBM cell lines. PRpS400 localization in cell nuclei. The treatment with Ki16425 (2.5 µM) inhibited the effect of LPA (100 nM) on the phosphorylation of PR at S400 at 15 min of stimulation in the GBM cell lines (**A**) U251, (**B**) LN229. PRpS400 (Green puncta), Hoechst (Blue). Representative images from 3 independent experiments. The photographs were taken at 60× magnification.

**Figure 3 cells-10-00807-f003:**
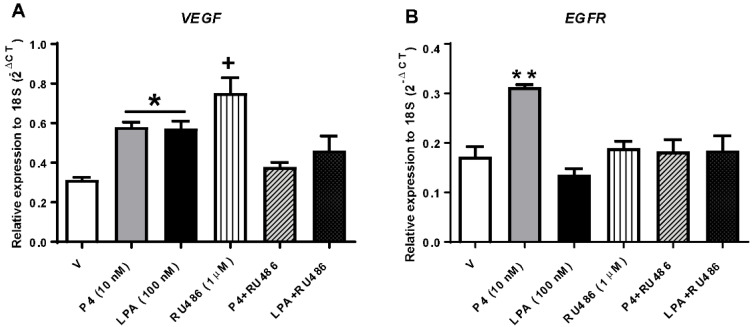
LPA modulates *VEGF* but not *EGFR* mRNA expression levels in GBM cells. U251 cells were treated with P4, LPA, RU486 and the combination of these treatments: P4 + RU486, LPA + RU486, or V (Vehicle; 0.1% ethanol) for 24 h, subsequently the gene expression of (**A**) *VEGF* and (**B**) *EGFR* was determined by RT-qPCR. Each column denotes the mean ± S.E.M. of each treatment, *n* = 4. The one-way ANOVA statistical test followed by a Tukey post-test determined the statistical difference * *p* < 0.02 P4 and LPA vs. V; + *p* < 0.009 vs. V and the combined treatments; ** *p* < 0.01 P4 vs. all other treatments.

**Figure 4 cells-10-00807-f004:**
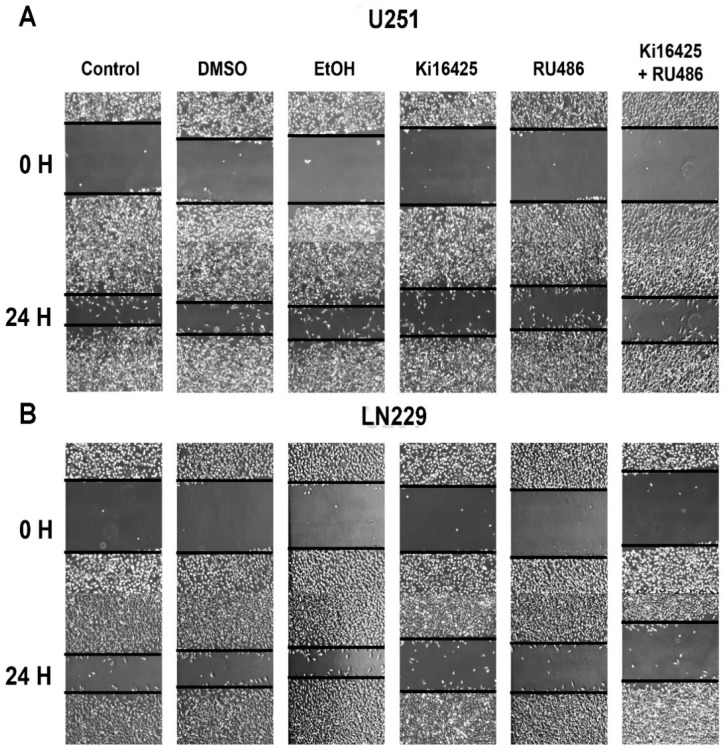
LPA/PR pathway signaling induces migration in GBM cells. (**A**) U251 and (**B**) LN299 cells were incubated for 24 h with complete medium (10% FBS), vehicles (DMSO and EtOH) and complete medium with Ki16425 (2.5 µM), RU486 (1 µM), or its combination. FBS was used as an LPA source. A total of 4 sections of the wound assay were analyzed per experiment and treatment. Representative images of two independent experiments are shown.

**Figure 5 cells-10-00807-f005:**
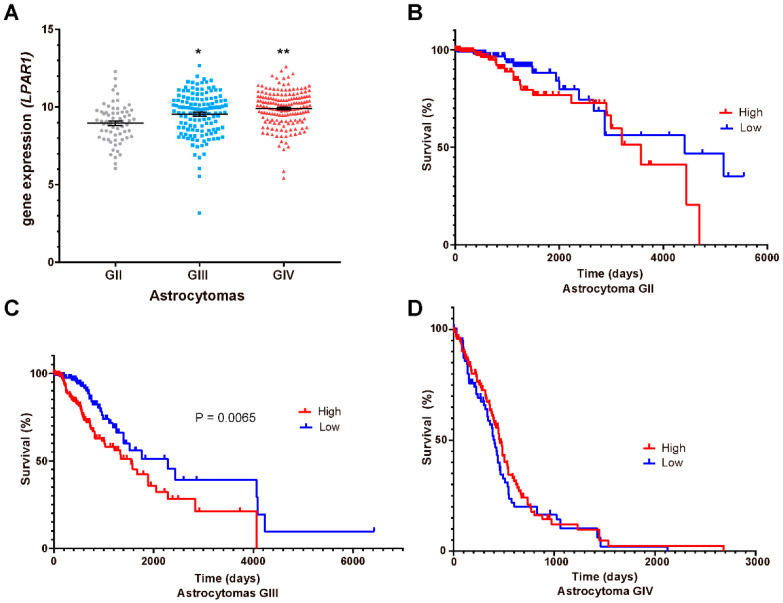
*LPAR1* mRNA expression and astrocytoma patient’s survival. (**A**) *LPAR1* mRNA expression (RNA-seq Illumina HiSeq, log2(norm count + 1)) in TCGA datasets ± SEM in different grades (G) of astrocytomas. The one-way ANOVA statistical test followed by a Tukey post-test determined the statistical difference * *p* < 0.05 GIII vs. GII and GIV, ** *p* < 0.05 GIV vs. GII and GIII. The Kaplan–Meier graphs show survival curves in patients that express low or high levels of *LPAR1* in astrocytomas (**B**) GII, (**C**) GIII and (**D**) GIV (GBM). Statistical analysis between high and low expression of *LPAR1* was performed with a Log-rank (Mantel–Cox) test. A value *p* < 0.05 was considered statistically significant, (**B**,**D**) were non-significant.

**Figure 6 cells-10-00807-f006:**
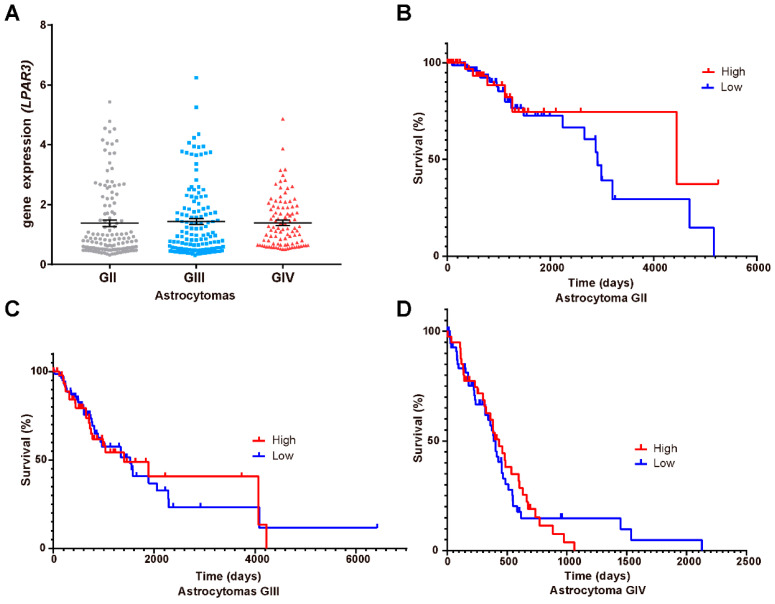
*LPAR3* mRNA expression and astrocytoma patient’s survival. (**A**) *LPAR3* mRNA expression (RNA-seq Illumina HiSeq, log2(norm count + 1)) in TCGA datasets ± SEM in different grades (G) of astrocytomas. The one-way ANOVA statistical test followed by a Tukey post-test determined the statistical difference. The Kaplan–Meier graphs show survival curves in patients that express low or high levels of *LPAR3* in astrocytomas (**B**) GII, (**C**) GIII and (**D**) GIV(GBM). Statistical analysis between high and low expression of *LPAR3* was performed with a Log-rank (Mantel–Cox) test. No statistical difference was observed.

**Figure 7 cells-10-00807-f007:**
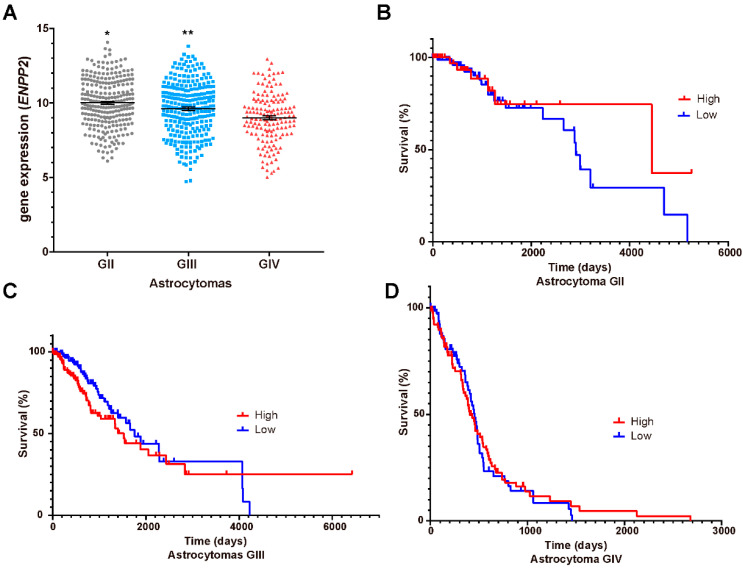
*ENPP2* mRNA expression and astrocytoma patient’s survival. (**A**) *ENPP2* mRNA expression (RNA-seq Illumina HiSeq, log2(norm count + 1)) in TCGA datasets ± SEM in different grades (G) of astrocytomas. The one-way ANOVA statistical test followed by a Tukey post-test determined the statistical difference. * *p* < 0.05 GII vs. GIII and GIV, ** *p* < 0.05 GIII vs. GII and GIV. The Kaplan–Meier graphs show survival curves in patients that express low or high levels of *ENPP2* in astrocytomas (**B**) GII, (**C**) GIII and (**D**) GIV (GBM). Statistical analysis between high and low expression of *ENPP2* was performed with a Log-rank (Mantel–Cox) test. No statistical difference was observed.

**Figure 8 cells-10-00807-f008:**
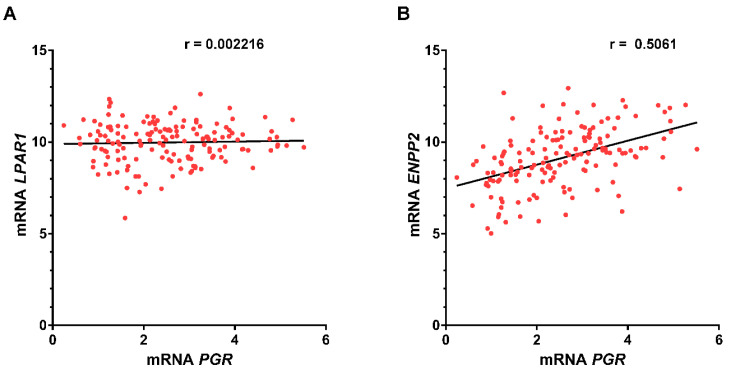
Correlation between *LPAR1/PGR* and *ENPP2/PGR* expression in GBM. The Spearman correlation coefficient was calculated to measure the association between (**A**) *LPAR1/PGR* and (**B**) *ENPP2/PGR* in GBM samples from the TCGA database.

**Figure 9 cells-10-00807-f009:**
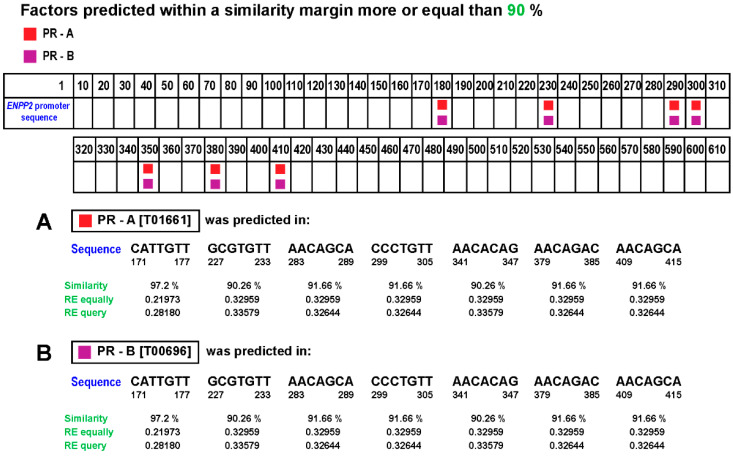
Possible binding sites for PR receptor in the *ENPP2* promoter sequence. Seven potential sequences were found for (**A**) PR isoform A and (**B**) PR isoform B in the *ENPP2* promoter.

## Data Availability

The data presented in this study for survival curves are openly available in: TCGA: low-grade astrocytomas: URL: https://portal.gdc.cancer.gov/projects/TCGA-LGG accessed on 5 March 2021, and high-grade astrocytomas: URL: https://portal.gdc.cancer.gov/projects/TCGA-GBM, accessed on 18 March 2021. Promotor sequence obtained from URL: https://www.ensembl.org/Homo_sapiens/Gene/Sequence?db=core;g=ENSG00000136960;r=8:119557086-119673453, accessed on 5 March 2021.

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
