# Peer review of "LPA1 Receptor Promotes Progesterone Receptor Phosphorylation through PKCα in Human Glioblastoma Cells"

_cells, 2021, doi:10.3390/cells10040807_

Round 1
Reviewer 1 Report
This interesting work shows that LPAR1 stimulation induces PR phosphorylation via PKCa, leading to a nuclear PRpS400 accumulation. Through this mechanism VEGF is upregulated and cell migration increased. LPAR1 und PR mRNA upregulation correlates with poorer survival in a TCGA dataset. The results presented are highly relevant for understanding GB pathophysiology. Therefore, the manuscript will be of high interest to many readers. The linguistic style is very good, making the manuscript easy to read and understand. Therefore, I support publication in principle.
One issue must be changed however: In the TCGA dataset, the authors compare grade II to grade III and grade IV astrocytoma. This comparison is highly problematic, as these tumors harbour very different molecular abberations. Please add to the existing data (or if you prefer replace the existing data) with TCGA data from with in one WHO grade subgroup. I.e. compare only grade IV tumors, and seperately only grade III tumors, and seperately only grade II tumors.
Reviewer 2 Report
This study conducts a series of in vitro studies to examine LPA1 receptor promotion via PKCa of progesterone receptor phosphorylation, leading to VEGF transcription, and potentially disease progression.
There are several issues that should be addressed:
1) The authors should discuss LPA production by ATX in the introduction, and give some supporting references. ENPP2/ATX does not appear til section 3.6 where it is introduced as part of the results discussion. This is not a typical format. Further, in this first paragraph of section 3.6, no citations are given for their references to the ATX discussion.
2) The authors should demonstrate mRNA expression levels of their cell lines used. They should demonstrate the their effects are not being mediated by LPA3 receptors.
3) All in vitro experiments should be conducted in at least two cell lines - this pertains particularly to Figure 1 and Figure 3, where just U251 cells were used.
4) Regarding Figure 5, similar curves should be generated with LPA3 and ENPP2 to demonstrate their argument that this is an LPA1 mediated result (these additional results can go in the supplemental.
5) Regarding the final paragraph in the results - these findings must be shown if a statement is to be made (ENPP2 PRE promoter sites).
6) There needs to be nomenclature consistency - LPA1, LPAR1, etc have been used throughout as an example. Line 436 - ENNP2 - note the error.
7) Proof of cell-line validation is standard for publication and must be included with the paper.
Round 2
Reviewer 1 Report
All suggestions for improvement have been solved. Now I strongly support publication. No further changes are necessary.
Reviewer 2 Report
Thank you for your changes to the paper.